# Identifying the Relationship between the Korean Medicine and Western Medicine in Factors Affecting Medical Service Use

**DOI:** 10.3390/healthcare10091697

**Published:** 2022-09-05

**Authors:** Young-eun Choi, Chul-woung Kim

**Affiliations:** 1Clinical Research Coordinating Team, Korea Institute of Oriental Medicine, Daejeon 34054, Korea; 2Department of Korean Medicine, The Graduate School, Pusan National University, Pusan 50612, Korea; 3Department of Preventive Medicine, Research Institute for Medical Sciences, College of Medicine, Chungnam National University, Daejeon 35015, Korea

**Keywords:** national study, medical service use, western medicine and Korean medicine, collaborative treatment, Korea Medical Panel Survey, integrative medicine

## Abstract

This study was conducted using data from the Korea Medical Panel Survey (KMPS) carried out in 2015. Importantly, the purpose of this study was to investigate the relationship between Korean medicine (KM) and Western medicine (WM) in medical service use. The general characteristics and the frequency of utilization of medical services were analyzed for 18,130 participants. Chi-square analysis was used to examine the factors that affected medical service use. Additionally, logistic regression analysis was conducted to examine the odds ratio (OR) between the KM Use with WM Use and KM&WM Use with disease group. The proportion of respondents who used KM&WM was the highest among those over 65 years of age and it was found to be statistically significant (*p* < 0.001). The OR for using KM and WM was 3.236 and it was also statistically significant (*p* < 0.001). Further, the ORs of KM&WM Use for all seven disease groups were greater than 1 and were statistically significant (*p* < 0.001) except for respiratory disease. The significant ORs of KM&WM Use were 10.342 (musculoskeletal), 2.073 (exogenous causes), 1.988 (nervous), 1.677 (digestive), 1.541 (circulatory) and 1.386 (skin). The findings in this study were attributed to a combination of social aspects such as the increasing incidence of chronic diseases among the elderly population, policy aspects such as the collaborative pilot project to promote collaborative treatment (CT), economic aspects, such as a lower total cost for CT and scientific aspects such as evidence supporting the efficacy of CT.

## 1. Introduction

The Korean medical system is a dual medical system in which both Korean medicine (KM) and Western medicine (WM), based on biomedical medicine, coexist. KM originated in Korea and has been continuously developed as a traditional medicine KM is composed mainly of acupuncture and herb medicine, treatments such as meditation and qigong, massage and bone setting. WM was introduced in Korea at the end of the 19th century and has supplanted KM in terms of health care.

The government recognizes both medical systems as accredited systems and the coexistence of these two systems gives the public the ability to select or use a preferred medical system [1]. In addition, the existence of an independent system suggests the need for collaborative treatment (CT) using WM and KM cooperatively.

On the other hand, due to social changes such as population aging, the incidence of chronic diseases among the elderly has increased. Moreover, the use of and interest in KM and CT combining KM and WM are growing and they are effective against chronic diseases [2].

For example, the government recognized the growing demands for KM and CT and, therefore, implemented various policies in 2010, such as standardizing the medical guidelines for KM and expanding the application of health insurance coverage to Korean oriental medical treatments [2]. In addition, the government’s support and policies permitting cooperative medical treatments combining KM and WM, including a national push for convergence and multidisciplinary treatments such as using CT, consultations, medical convergence and integrated medicine, are continuously reviewed to promote the simultaneous use of both medical systems [3].

To establish and activate policies to promote the use of oriental medicine, as well as policies related to the simultaneous use of KM and WM, it is necessary to identify and understand actual medical use and the needs of the public [4,5,6,7,8,9]. For this reason, it is necessary to conduct a study based on large-scale social survey data reflecting real-world usage to determine whether the use of KM and WM is complementary and/or alternative.

So far, various studies have been conducted to analyze the patterns of KM usage [10,11,12,13]. According to the results of a previous study, KM use has a complementary and substitute relationship with WM for diseases related to the respiratory system, skin and subcutaneous tissue, musculoskeletal system, injuries, poisoning and exogenous diseases [1]. However, since the study was performed using relatively old social survey data, there are limitations in the identification of factors related to the current use of KM and WM. Therefore, this recent study was performed to analyze the affecting factors comparing KM use and WM use based on data from a recent Korea Medical Panel Survey (KMPS) carried out in 2015.

In the analysis of medical service use, it is difficult to explain using only a single factor, because medical service use is represented by various factors. Therefore, this study analyzed the relations between control variables related to the use of KM and WM by the Andersen behavior model and then analyzed the relationship between KM use and WM use [14,15]. This model has been widely used to analyze factors that induce medical use in the research [16]. Accordingly, using Andersen’s behavioral model as a control variable, a complementary relationship analysis between KM use and WM use was conducted. In addition, an analysis of the complementary relationship between KM use and WM use was performed by disease.

In Korea, where KM and WM coexist, the government must establish a system that allows individuals to use appropriate medical services. Therefore, this study aims to identify appropriate medical use by analyzing the complementary relationship between KM and WM. Based on this study, we suggest health policies and develop and promote systems that can improve health care efficiency.

## 2. Materials and Methods

### 2.1. Research Subjects

This study was conducted using KMPS 2015 annual aggregate data. The annual combined data for 2015 was collected from 1 January 2015 to 31 December 2015 and a total of 18,130 people from 6607 households were included. Respondents have a total of 300,205 outpatient medical use records and, in this study, all outpatient medical users were analyzed to confirm the relationship between KM and WM in all outpatient medical use.

### 2.2. Variables Used in the Study

#### 2.2.1. The Status of Medical Service Use

The variables for medical service in the descriptive statistics were KM and WM, KM only, WM only, and KM and WM. The KM variable was classified into herbal medicine (HM) and Non-HM. The WM variable included medical services at university hospitals, general hospitals and private clinics, whereas the KM variable included medical services at KM hospitals and KM clinics. The KM use variable was classified into total KM use, HM use and Non-HM use. HM was defined to include packed HM, decoction and restorative HM, whereas Non-HM was defined to include acupuncture, moxibustion, cupping, manual therapy, physical therapy, etc. The variables selected for visitation frequency and medical expenses were mean visitation frequency, mean medical expenses and mean medical expenses per visit. 

#### 2.2.2. Factors Related to Medical Service Use

In the relationships between WM Use and KM Use, the dependent variable was WM use and the independent variables were KM use, HM Use and Non-HM Use. In the relationship between KM&WM Use and diseases, the dependent variable was KM&WM use and the independent variables were nervous system diseases (NSD), circulatory system diseases (CSD), respiratory system diseases (RSD), digestive system diseases (DSD), skin and subcutaneous tissue diseases (SSD), musculoskeletal system diseases (MSD) and injury, poisoning and exogenous causes diseases (IPED). As stated in Andersen’s medical model, the control variables were sex, age, education, residential area, household income, insurance, exercise problem, anxiety/depression, disability and chronic disease (Table 1).

### 2.3. Statistical Analysis

Based on the specific purposes, statistical methods were used for analysis. The program used for all statistical analyses was SPSS WIN 18.0 and the significance level of the statistical test was 5%.

#### 2.3.1. General Characteristics 

Frequency analysis was performed for the general characteristics for seven disease categories and the number and ratio were determined.

#### 2.3.2. Current Status of Medical Service Use

Frequency analysis was conducted to identify the status of medical service use. The number and ratio of medical service use were determined for outpatients, KM and WM, KM only, WM only, KM (total HM and Non-HM) and WM. The mean and standard deviation (SD) of the usage frequency, medical expenses and medical expenses per visit were determined for KM, HM, Non-HM and WM.

#### 2.3.3. Frequency of Medical Service Uses according to Disease

Frequency analysis of the number of medical service uses for the treatment of seven common diseases was performed. After selecting the cases by dividing the subjects according to the disease, the number and ratio were calculated for respondents who selected total, WM, KM, KM and WM, WM only, and KM only.

#### 2.3.4. Analysis of Related Factors for KM Use and WM Use

Logistic regression analysis was conducted on the relationship between the use of WM and KM. WM use was set as the dependent variable, whereas KM use was set as the independent variable. The regression coefficients and statistical significance were used to confirm the relations. An odds ratio (OR) greater than 1 indicated a positive relationship (+) and complementary usage, whereas a ratio value of less than 1 but greater than 0 indicated a negative relationship (−) and substitute usage. The KM variable was classified into KM Use (Model I), HM Use (Model II) and Non-HM Use (Model III).

#### 2.3.5. Analysis of Related Factors for Medical Service Use according to Disease

After selecting cases using KM or WM, logistic regression was used to analyze the relationship between the use of KM&WM and seven common diseases. KM&WM Use was set as the dependent variable, whereas diseases were set as the independent variables. The disease variables were divided into NSD (Model I), RSD (Model II), CSD (Model III), DSD (Model IV), SSD (Model V), MSD (Model VI) and IPED (Model VII). The interpretation method of OR was the same as that described in Section 2.3.4.

## 3. Results

### 3.1. General Characteristics of Respondents

The number of respondents with chronic diseases was 10,089 (55.6%), which was more than those without chronic diseases. The disease group with the highest frequency of medical use was RSD with 10,294 (56.8%) respondents, followed by DSD with 6950 (38.3%) respondents, MSD with 6289 (34.7%) respondents, CSD with 4063 (22.4%) respondents, SSD with 2986 (16.5%) respondents, IPED with 2440 (13.5%) respondents and NSD with 1257 (6.9%) respondents (Table 2).

### 3.2. Status of KM and WM Service Use 

The total number of respondents was 18,130 (100%), of which 15,608 (86.1%) were outpatients. In total, 2744 (15.1%) respondents used KM, whereas 5073 (83.1%) used WM, 2632 (14.5%) used KM and WM, 112 (0.6%) used KM only and 12,442 (68.6%) used WM only. Among the respondents who used KM, a larger proportion reported using non-HM, with 916 (33.4%) using HM and 2563 (93.4%) using Non-HM. The mean number of visits was 10.02 times for KM, 2.72 times for HM, 10.60 times for Non-HM and 16.19 times for WM. The mean medical expense was 151,677.13 won for KM, 293,552.22 won for HM, 56,727.14 won for Non-HM and 233,543.52 won for WM (Table 3, Figure 1).

### 3.3. Analysis of Characteristics of Factors Related to Medical Service Use

Respondents under the age of 45 years accounted for the largest proportion of users of KM only and WM only as well as of non-users, whereas those over 65 years of age accounted for the highest proportion of users of KM and WM, which was statistically significant (*p* < 0.001). A larger proportion of users and non-users of all medical services selected “No” in response to questions about exercise problems, anxiety/depression and disability as need factors and the results were statistically significant (*p* < 0.001). However, a larger number of respondents with chronic diseases selected “Yes” for KM and WM as well as WM only, which was also statistically significant (Table 4, *p* < 0.001).

### 3.4. Analysis of Related Factors for Medical Service Use

The statistical results for Model I were significant (*p* < 0.001) and this model had an explanatory power of 27.7%. The OR for KM Use was 3.236 with a 95% confidence interval (CI) of 2.599–4.03 and the results were statistically significant (Table 5, *p* < 0.001).

The statistical results for Model II were also significant (*p* < 0.001) and this model had an explanatory power of 26.5%. The OR for HM Use was 2.533 (CI = 1.744–3.648) and the results were statistically significant (Table 5, *p* < 0.001).

The statistical results for Model III were statistically significant (*p* < 0.001) and this model had an explanatory power of 27.8%. The OR for Non-HM Use was 3.670 (CI = 2.891–4.660) and the result was statistically significant (Table 5, *p* < 0.001).

### 3.5. Analysis of Related Factors for Medical Service Use according to Disease

For NSD, the OR of KM&WM Use (OR = 1.988, *p* < 0.001) was greater than 1, indicating a positive (+) relationship and statistical significance. For RSD, the ORs of KM&WM Use (OR = 1.025, *p* > 0.05) were greater than 1, but they were not statistically significant. As described earlier, the ORs of KM&WM Use for all seven disease groups were greater than 1 and were statistically significant except for RSD. The significant ORs of KM&WM Use are 10.342 (MSD), 2.073 (IPED), 1.988 (NSD), 1.677 (DSD), 1.541 (CSD) and 1.386 (SSD) (Table 6).

## 4. Discussion

The present study investigated the use of outpatient medical services based on the annual integrated data collected from KMPS performed in 2015, which is the recent large-scale national social survey data. In addition, the study identified the changes in medical service usage and the patterns of medical expenditure for KM and WM in comparison to a previous study [1]. The comparison results of this study and the previous study are as follows.

The general characteristics of the respondents were similar in both studies. The number of respondents in 2015 (18,130) included in the current study was slightly lower than the number in 2009 (19,413) included in the previous study. Additionally, the number of outpatients in 2015 (15,608, 86.1%) was slightly greater than the number in 2009 (15,095, 77.8%) [1]. Furthermore, the number of respondents who used medical services increased overall, excluding those that used KM only.

Frequency analysis showed that the mean visitation frequency for all medical services in 2007 and 2015 increased overall (KM—7.21 to 10.02, HM—1.57 to 2.72, Non-HM—8.15 to 10.60 and WM—13.16 to 16.19). This is consistent with the increase in the annual average visits by outpatients in the OECD report, from 11.8 in 2007 to 14.6 in 2015 [17]. In addition, the medical expenses increased in 2015, in line with the information in the 2015 OECD report [17]. The increase in medical expenses was different for the medical services. The medical service with the largest increase in medical expenses was WM, with a mean expense of 97,166.54 won in 2009 and 233,543.52 won in 2015. The mean medical expense increased from 194,211.15 won to 293,552.22 won for HM, from 118,306.75 won to 151,677.13 won for KM and from 54,547.72 won to 56,727.14 won for Non-HM [1].

Compared with the results in 2009, the characteristics of factors related to medical service use in 2015 were determined to be different. Respondents under 45 years of age comprised the highest proportion of users of medical services (KM, WM and KM&WM) in 2009, whereas those over 65 years of age comprised the highest proportion of KM&WM users in 2015. It could be predicted that KM and KM&WM increased in elderly individuals because KM and KM&WM were effective for the treatment of chronic diseases [2]. The number of people in the chronic diseases group also increased with all medical users and non-users. This is consistent with the increase in the incidence of chronic diseases caused by the aging of the population, which is believed to be the reason for the increase in the number of KM&WM users over 65 years of age [2,18].

In 2009 and 2015, the ORs for KM Use, HM Use and Non-HM Use of WM Use were greater than 1. The OR for KM Use increased from 2.946 in 2009 to 3.236 in 2015, the OR for HM use decreased from 2.703 in 2009 to 2.523 in 2015 and the OR for Non-HM use increased from 2.915 in 2009 to 3.670 in 2015 [1]. This means that an increased number of patients using KM or Non-HM was also more likely to use WM, compared to those who did not. It was predicted since HM did not include insurance claims, the number of times that the HM was ultimately used was small and the effect on its relationship with WM was lower [19].

Analysis results of the ORs for seven diseases related to KM&WM Use showed that ORs of all diseases were greater than 1. The diseases that indicated statistical significance were NSD, CSD, DSD, SSD, MSD and IEPD. The highest OR was 10.342 for MSD and the following ORs were 2.073 for IEPD, 1.988 for NSD, 1.677 for DSD, 1.541 for CSD, 1.386 for SSD. Specifically, when the number of respondents with MSD was higher, the number of respondents using KM&WM was 10.342 times more than that of the respondents without MSD. Conclusively, among the people who use KM&WM, most had MSD disease.

This study indicated that the main reason for these findings and the differences from the results of the previous study was the recent increase in the use of KM and KM&WM as CT among elderly individuals [20,21,22]. Therefore, we investigated the social, policies, economic and scientific causes based on the above research results.

First, in terms of the social aspects, disease patterns change as the elderly population increases worldwide; therefore, the use of complementary and alternative medicine (CAM), such as Korean oriental medicine, has increased due to its improved efficacy for the treatment of chronic diseases [23]. MSD are caused by diseases such as osteoarthritis, rheumatoid arthritis and osteoporosis in elderly individuals and is often treated with both KM and WM [14]. WM use mainly consists of surgery, drug treatments and physical therapy, whereas KM use consists of conservative treatments such as acupuncture, herbs and Chuna [24,25].

Second, in terms of policy aspects, as Korean–Western Integrative Medicine (KWIM) treatment was recently established as a national policy, corresponding national projects led to an increase in effect, economic efficiency and satisfaction. One of these projects, the “first-stage pilot project for KM-WM collaborative treatments,” was part of the third Korean Medicine Development Plan (2016–2020) [26]. According to a recent report regarding this pilot project, the use of CT increased and the most frequent diseases treated with CT were MSD and NSD [27]. In the same report, MSD accounted for 42.29% of the total number of insurance claims for CT, whereas NSD and IEPD accounted for 28.13% and 16.75%, respectively.

Third, from an economic point of view, the total medical expenses for CT were lower than those for KM or WM, which is attributed to the economic benefits [3,18]. A recent report indicated that CT users had a lower hospital visitation frequency and longer intervals in between the visits. Therefore, although the cost of each visit (cost per unit) increased, the number of medical uses (treatment period) decreased. Moreover, in comparison to previous treatments using only one medical system, the total medical cost of CT was lower. In other words, the total economic benefits to patients can be increased by using both KM and WM as CT, thereby reducing the burden of medical costs to patients [3].

Finally, from a scientific aspect, the effectiveness of KM is increasingly demonstrated with scientific evidence. Consequently, the public interest in CT has also increased [20]. A recent study on CT based on an analysis of 24 case reports indicated that diseases of the nervous system were the most likely to be treated with CT due to its excellent treatment effects [28]. In 2016, the diseases that received the most consultation for CT were musculoskeletal system, nervous system and exogenous diseases [29]. However, the limitation of this finding was that it only included consultations for acupuncture medicine. Although the validation of the medical effect of CT in several medical fields is incomplete, the efficacy of Non-HM for disease is gradually being scientifically proven [21,22].

In conclusion, the number of elderly patients suffering from chronic diseases has increased in recent years. The limitations of surgical treatment are obvious and therefore, conservative treatments are regarded positively. As a result, the use of oriental medicine and CT has increased and various government support policies are being implemented. However, in the Korean medical service environment where KM and WM coexist as independent systems, it is difficult to resolve conflicts of interest and conflicts between two overlapping medical fields. Consequently, we analyzed the patterns of medical service use among outpatients to provide a foundation for resolving this conflict. We found an increase in the use of KM and WM for the treatment of chronic, nervous and MSD in elderly patients. Accordingly, we analyzed the reasons for these results multilaterally. Furthermore, we suggest that the government should resolve the conflicts between the two medical systems by developing integrated health policies to achieve positive outcomes.

## 5. Limitations and Future Research

First, the KMPS is not intended to identify factors related to the use of outpatient medical services in KM and WM; therefore, there are limitations in the selection of related variables. Second, since this study was a cross-sectional study, various biases that occur during the data collection process may be reflected in the data. This study used KMPD data in 2015 and a time series analysis study using data from 2015 to 2018 is planned. Third, the results cannot be generalized to other countries because the cultural background and compensation system of other countries are different. In order to solve this limitation, it would be useful to reconfirm the factors affecting medical use by adding a question to understand why respondents choose KM and WM and their relationship.

Despite these limitations, the survey dataset provides a representative sample based on many populations. This study can provide a resource for clinicians, researchers and policy makers who need to confirm the current status of KM and WM use and the relationship between both medical services and to understand access to medical service when using outpatient medicine in Korea. The major advantage is that the well-known Andersen behavior model can be used to analyze medical use for a large number of respondents.

## 6. Conclusions

Among patients using outpatient treatment, the ratio of using KM and WM is high in the elderly and among sufferers of chronic diseases.

A complementary relationship between KM and WM was confirmed in outpatient treatment.

For MSD, IEPD and NSD diseases, there is a high correlation between the use of KM and WM when using outpatient medical care.

This study aims to provide a foundation for promoting recently established health care policies in support of Korean–Western CT.

However, further research should be conducted to analyze medical use in future years.

## Figures and Tables

**Figure 1 healthcare-10-01697-f001:**
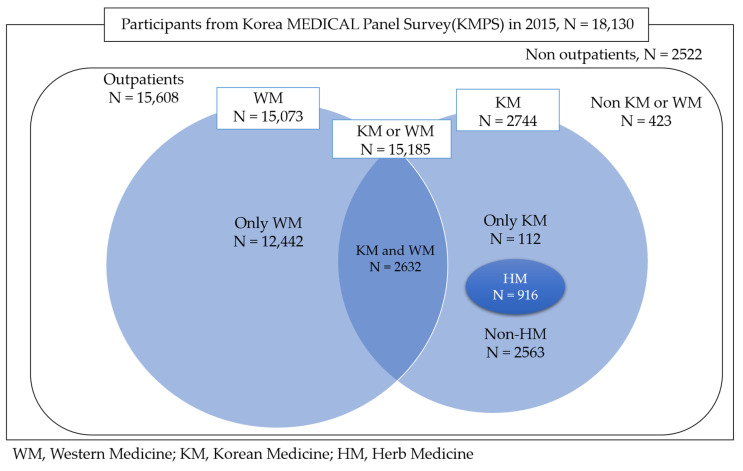
Classification of KM use and WM use by outpatient medical users.

**Table 1 healthcare-10-01697-t001:** Variables of the Factors Related to Medical Service Use.

Variable		Options
**Independent variable**	WM use	0: No, 1: Yes
	KM&WM Use	0: No, 1: Yes
**Dependent variable**	KM use	0: No, 1: Yes
	HM Use	0: No, 1: Yes
	Non-HM Use	0: No, 1: Yes
	NSD	0: No, 1: Yes
	CSD	0: No, 1: Yes
	RSD	0: No, 1: Yes
	DSD	0: No, 1: Yes
	SSD	0: No, 1: Yes
	MSD	0: No, 1: Yes
	IPED	0: No, 1: Yes
**Control variable**	Sex	0: Male, 1: Female
	Age	0: Under 45 years, 1: Over 45∼Under 55, 2: Over 55∼Under 65, 3: Over 65
	Education	0: Under the middle school, 1: Over the high school
	Residential Area	0: Outside metropolitan area, 1: Metropolitan city
	Household Income (10,000 won)	0: Under 3500, 1: Over 3500
	Insurance	0: No, 1: Yes
	Exercise Problems	0: No, 1: Yes
	Anxiety/Depression	0: No, 1: Yes
	Disability	0: No, 1: Yes
	Chronic Disease	0: No, 1: Yes

WM, Western Medicine; KM, Korean Medicine; HM, Herb Medicine; NSD, Nervous System Diseases; CSD, Circulatory System Diseases; RSD, Respiratory system Diseases; DSD, Digestive System Diseases; SSD, Skin and Subcutaneous tissue Diseases; MSD, Musculoskeletal System Diseases; IPED, Injury, Poisoning and Exogenous causes Diseases.

**Table 2 healthcare-10-01697-t002:** General Characteristic of Respondents (N = 18,130, All subjects).

Variables		*n*	%
**Sex**	Male	8726	48.1
	Female	9404	51.9
**Age (years)**	Under 45	8534	47.1
	45–54	2814	15.5
	55–64	2533	14
	Over 65	4249	23.4
**Education**	Below middle school	7140	39.4
	Above high school	10,990	60.6
**Residential Area**	Outside metropolitan area	10,502	57.9
	Metropolitan city	7628	42.1
**Household Income (10,000 won)**	Under 3500	7104	39.2
	Over 3500	11,026	60.8
**Insurance**	Yes	18,027	99.42
**Exercise Problem (n = 13,433)**	Yes	1727	12.9
**Anxiety/Depression (n = 13,433)**	Yes	1739	12.9
**Disability**	Yes	1073	5.9
**Chronic Disease**	Yes	10,089	55.6
**NSD**	Yes	1257	6.9
**CSD**	Yes	4063	22.4
**RSD**	Yes	10,294	56.8
**DSD**	Yes	6950	38.3
**SSD**	Yes	2986	16.5
**MSD**	Yes	6289	34.7
**IPED**	Yes	2440	13.5
**Total**		18,130	100.0

NSD, Nervous System Diseases; CSD, Circulatory System Diseases; RSD, Respiratory system Diseases; DSD, Digestive System Diseases; SSD, Skin and Subcutaneous tissue Diseases; MSD, Musculoskeletal System Diseases; IPED, Injury, Poisoning and Exogenous causes Diseases.

**Table 3 healthcare-10-01697-t003:** Medical Service Use by Respondents (*n* = 18,130, All subjects).

Variables	*n* or Mean	% or ±SD
**Respondents**		
**Outpatients**	15,608	86.1
**KM or WM**	15,185	83.8
**KM and WM**	2632	14.5
**KM**	2744	15.1
**WM**	15,073	83.1
**KM Only**	112	0.6
**WM Only**	12,442	68.6
**Total**	18,130	100.0
**KM**		
**HM**	916	33.4
**Non-HM**	2563	93.4
**Total**	2744	100.0
**Mean Visitation Frequency (times)**		
**KM**	10.02	±16.25
**HM**	2.72	±5.14
**Non-HM**	10.60	±16.68
**WM**	16.19	±21.73
**Mean Medical expenses (won)**		
**KM**	151,677.13	±330,164.52
**HM**	293,552.22	±357,439.65
**Non-HM**	56,727.14	±162,173.13
**WM**	233,543.52	±498,705.18
**Mean Medical expenses per visit (won)**		
**KM**	15,137.44	
**HM**	107,923.61	
**Non-HM**	5351.62	
**WM**	14,425.17	

WM, Western Medicine; KM, Korean Medicine; HM, Herb Medicine.

**Table 4 healthcare-10-01697-t004:** Chi-square Analysis of KM and WM Service Use (*n* = 18,130, All subjects).

Variables		Medical Service Use			Non Use	X^2^
		KM only	KM & WM	WM only		
		*n* (%)	*n* (%)	*n* (%)	*n* (%)	
**Predisposing Factors**						
**Sex**	**Male**	53 (47.30)	900 (34.20)	5840 (46.90)	1933 (65.70)	574.19 ***
	**Female**	59 (52.70)	1732 (65.80)	6602 (53.10)	1011 (34.30)	
**Age (years)**	**Under 45**	53 (47.30)	672 (25.50)	5853 (47.00)	1956 (66.40)	1305.68 ***
	**45–54**	36 (32.10)	410 (15.60)	1826 (14.70)	542 (18.40)	
	**55–64**	14 (12.50)	525 (19.90)	1743 (14.00)	251 (8.50)	
	**Over 65**	9 (8.00)	1025 (38.90)	3020 (24.30)	192 (6.60)	
**Education**	**Below middle school**	20 (17.90)	1252 (47.60)	5414 (43.50)	454 (15.40)	892.61 ***
	**Above high school**	92 (82.10)	1380 (52.40)	7028 (56.50)	2490 (84.60)	
**Enabling Factors**						
**Residential Area**	**Outside metropolitan area**	65 (58.00)	1514 (57.50)	7313 (58.80)	1610 (54.70)	16.54 ***
	**Metropolitan city**	47 (42.00)	1118 (42.50)	5129 (41.20)	1334 (45.30)	
**Household** **Income (10,000 won)**	**Under 3500**	31 (27.70)	1292 (49.10)	4868 (39.10)	913 (31.00)	197.07 ***
	**Over 3500**	81 (72.30)	1340 (50.90)	7574 (60.90)	2031 (69.00)	
**Insurance**	**No**	0 (0.00)	15 (0.60)	70 (0.60)	18 (0.60)	0.74
	**Yes**	112 (100.00)	2632 (99.40)	12,442 (99.40)	2944 (99.40)	
**Need Factors**						
**Exercise Problems**	**No**	93 (96.90)	1845 (79.40)	7843 (86.80)	1925 (97.50)	320.75 ***
	**Yes**	3 (3.10)	478 (20.60)	1196 (13.20)	50 (2.50)	
**Anxiety/Depression**	**No**	87 (90.60)	1923 (82.80)	7837 (86.70)	1847 (93.50)	112.96 ***
	**Yes**	9 (9.40)	400 (17.20)	1202 (13.30)	128 (6.50)	
**Disability**	**No**	111 (99.10)	2421 (92.00)	11,679 (93.90)	2846 (96.70)	62.37 ***
	**Yes**	1 (0.90)	211 (8.00)	763 (6.10)	98(3.30)	
**Chronic Disease**	**No**	74 (66.10)	572 (21.70)	5105 (41.00)	2290 (77.80)	1955.96 ***
	**Yes**	38 (33.90)	2060 (78.30)	7337 (59.00)	654 (22.20)	
**Total**		112 (100)	2632(100)	12,442 (100)	2944 (100)	

***: *p*-value <0.001, *p*-values were calculated using chi-squared test. KM, Korean Medicine; WM, Western Medicine.

**Table 5 healthcare-10-01697-t005:** Factors Related to Use the KM and WM *(n* = 13,433, selecting all subjects and including analysis cases).

Variable	Model I	Model II	Model III
	OR ^†^	OR ^†^	OR ^†^
**Dependent Variable**			
**WM Use**			
**Independent Variable**			
**KM Use (No)** **: Yes**	3.236 *** (2.599–4.03)		
**HM Use (No)** **: Yes**		2.523 *** (1.744–3.648)	
**Non-HM Use (No)** **: Yes**			3.670 *** (2.891–4.660)
**Control Variable**			
**Sex (Male): Female**	2.345 ***	2.406 ***	2.346 ***
**Age (Under 45 years):**			
**Over 45-Under 55**	1.21 **	1.242 ***	1.202 ***
**Over 55-Under 65**	1.908 ***	1.989 ***	1.893 ***
**Over 65**	3.448 ***	3.562 ***	3.427 ***
**Education (Below middle school)** **: Above high school**	1.142	1.140	1.145
**Residential Area (Outside** **metropolitan area): metropolitan city**	0.944	0.943	0.942
**Household Income (10,000 won)** **(Under 3500): Over 3500**	1.135 *	1.124	1.139 *
**Exercise Problem (No): Yes**	1.484 *	1.524 ***	1.484 *
**Anxiety/Depression (No): Yes**	0.983	0.988	0.983
**Disability (No): Yes**	1.28	1.268	1.282
**Chronic disease (No): Yes**	4.762 ***	4.9 ***	4.756 ***
***n* (Including analysis cases)**	13,433	13,433	13,433
**Chi-square**	2338.642 ***	2226.849 ***	2350.223 ***
**-2 log likelihood**	9213.893	9325.686	9202.311
**R Square ^‡^**	0.277	0.265	0.278

*: *p* < 0.05, **: *p* < 0.01, ***: *p* < 0.001, *p*-values were calculated using logistic regression analysis. ^†^: 95% confidence interval, comparing the outcome with WM Use. ^‡^: an explanatory power of the model. OR, Odds ratio; WM, Western Medicine; KM, Korean Medicine; HM, Herb Medicine.

**Table 6 healthcare-10-01697-t006:** Factors Related to KM&WM Use and Seven Diseases. *(n* = 11,457, Selected the KM or WM Use and including analysis cases).

Variable	Model I	Model II	Model III	Model IV	Model V	Model VI	Model VII
	OR ^†^	OR ^†^	OR ^†^	OR ^†^	OR ^†^	OR ^†^	OR ^†^
**Dependent Variable** ** ^†^ **							
**KM&WM Use**							
**Independent Variable**							
**NSD (No): Yes**	1.988 ***						
**RSD (No): Yes**		1.025					
**CSD (No): Yes**			1.541 ***				
**DSD (No): Yes**				1.677 ***			
**SSD (No): Yes**					1.386 ***		
**MSD (No): Yes**						10.342 ***	
**IPED (No): Yes**							2.073 ***
**Control Variable**							
**Sex (Male): Female**	1.698 ***	1.729 ***	1.673 ***	1.715 ***	1.728 ***	1.623 ***	1.728 ***
**Age (Under 45 years):**							
**Over 45-Under 55**	1.343 ***	1.356 ***	1.403 ***	1.321 ***	1.376 ***	0.97	1.334 ***
**Over 55-Under 65**	1.647 ***	1.678 ***	1.728 ***	1.578 ***	1.710 ***	1.017	1.657 ***
**Over 65**	1.660 ***	1.721 ***	1.744 ***	1.581 ***	1.748 ***	0.977	1.766 ***
**Education (Below middle school): Above high school**	1.014	1.008	1.024	1.000	1.009	1.232 **	1.001
**Residential Area (Outside metropolitan area): metropolitan city**	1.083	1.087	1.097	1.085	1.087	1.171 **	1.082
**Household Income (10,000 won) (Under 3500): Over 3500**	0.984	0.976	0.981	0.976	0.978	1.007	0.983
**Exercise Problem (No): Yes**	1.161 *	1.183 *	1.201 **	1.190 *	1.188 *	1.06	1.169 *
**Anxiety/Depression (No): Yes**	0.971	1.018	1.018	1.014	1.013	1.000	1.005
**Disability (No): Yes**	0.935	0.952	0.955	0.959	0.945	0.991	0.964
**Chronic disease (No): Yes**	1.559 ***	1.604 ***	1.579 ***	1.546 ***	1.588 ***	1.292 ***	1.613 ***
** *n* ** **(Including analysis cases)**	11,457	11,457	11,457	11,457	11,457	11,457	11,457
**Chi-square**	501.505 ***	410.241 ***	480.231 ***	524.556 ***	443.103 ***	1933.378 ***	553.934 ***
**2 log likelihood**	11,051.786	11,143.051	11,073.061	11,028.736	11,110.189	9619.914	10,999.358
**R Square ^‡^**	0.067	0.055	0.065	0.07	0.06	0.244	0.074

*: *p* < 0.05, **: *p* < 0.01, ***: *p* < 0.001, *p*-values were calculated using logistic regression analysis. ^†^: 95% confidence interval, comparing the outcome with KM&WM Use. ^‡^: an explanatory power of the model. OR, Odds ratio; WM, Western Medicine; KM, Korean Medicine; NSD, Nervous System Diseases; CSD, Circulatory System Diseases; RSD, Respiratory system Diseases; DSD, Digestive System Diseases; SSD, Skin and Subcutaneous tissue Diseases; MSD, Musculoskeletal System Diseases; IPED, Injury, Poisoning and Exogenous causes Diseases.

## Data Availability

The datasets used or analyzed during the current study are available from the corresponding author upon reasonable request.

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
