# Peer review of "Identifying the Relationship between the Korean Medicine and Western Medicine in Factors Affecting Medical Service Use"

_healthcare, 2022, doi:10.3390/healthcare10091697_

Round 1

Reviewer 1 Report

Thank you for the opportunity to review the manuscript “Identifying the Relationship Between the Korean Medicine and Western Medicine in Factors Affecting Medical Service Use” (healthcare-1860878).

The authors submitted a study to investigate the relationship between Korean medicine and Western medicine in medical service use. The general characteristics and the frequency of utilization of medical services were analysed for 18,130 participants in 2015.

However, parts of the paper must be carefully revised.

The introduction would be clearer if authors present definitions of key terms in a table especially WM, KM, CT. The authors should also use current literature to lead to the main part of the study. What are the hypotheses?

Line 56 the punctuation mark is missing.

Discussion

Line 236 “The comparison results of this study and the previous study are as follows.” Line 236 and 238 is the same text.

Overall, the discussion must be revised and linked to the results. It is important to discuss the results with more and recent literature.

Please insert a figure/table with main results. The data of this figure should aim on future implications.

Limitations

Essential limitation is the obsolete record. Data resource: 2015.

Author Response

Thank you for the opportunity to review the manuscript “Identifying the Relationship Between the Korean Medicine and Western Medicine in Factors Affecting Medical Service Use” (healthcare-1860878).

The authors submitted a study to investigate the relationship between Korean medicine and Western medicine in medical service use. The general characteristics and the frequency of utilization of medical services were analysed for 18,130 participants in 2015.

However, parts of the paper must be carefully revised.

The introduction would be clearer if authors present definitions of key terms in a table especially WM, KM, CT. -> Full names and definitions can be found in the overview and in the footnotes below the table.

The authors should also use current literature to lead to the main part of the study.

-> Added references(Please see the attachment).

What are the hypotheses?

-> Edited(Please see the attachment)

Line 56 the punctuation mark is missing.

-> Edited(Please see the attachment)

Discussion

Line 236 “The comparison results of this study and the previous study are as follows.” Line 236 and 238 is the same text.

-> Edited(Please see the attachment)

Overall, the discussion must be revised and linked to the results. It is important to discuss the results with more and recent literature.

-> There are not many up-to-date references using the KMPS. Once this paper is submitted, I would like to conduct additional research using KMPS data and secure relevant evidence.

Please insert a figure/table with main results. The data of this figure should aim on future implications.

-> Added a figure and table(as supplement)

Limitations

Essential limitation is the obsolete record. Data resource: 2015.

-> Edited(Please see the attachment)

Reviewer 2 Report

This study is expected to contribute to management decisions for chronic diseases among the elderly.

My comments are as follows:

1. The study was conducted in 2015. Why it took 7 years to report the findings? Will the findings still valid and reliable as a reference and be able to benefit the healthcare system?

2. To clarify study population (Patient and public subjects) in the methodology

3. Limitations and strength of the study should be further elaborated. eg. the results cannot be generalised outside of Korea because the cultural background and reimbursement systems of other countries are different. 

4. Most of the comparisons are made between 2009 (Reference #1) and 2015. The authors should include  a more diverse comparisons

eg  Kim, D., Lim, B. & Kim, C. Relationship between patient satisfaction with medical doctors and the use of traditional Korean medicine in Korea. BMC Complement Altern Med 15, 355 (2015). https://doi.org/10.1186/s12906-015-0879-x

Author Response

<Response for reviewer 2 >

This study is expected to contribute to management decisions for chronic diseases among the elderly.

My comments are as follows:

1. The study was conducted in 2015. Why it took 7 years to report the findings? Will the findings still valid and reliable as a reference and be able to benefit the healthcare system?

-> We started this study in 2018 and the latest annual data available at the beginning of 2018 was 2015. This study completed in 2020 after performing and writing data analysis for a year. It has been submitted to several journals since 2020, but withdrawn due to lack of reviewers. After this manuscript is submitted, detailed follow-up studies will be conducted using the latest data.

2. To clarify study population (Patient and public subjects) in the methodology

-> Please see the attachment

3. Limitations and strength of the study should be further elaborated. eg. the results cannot be generalised outside of Korea because the cultural background and reimbursement systems of other countries are different.

 -> Please see the attachment

4. Most of the comparisons are made between 2009 (Reference #1) and 2015. The authors should include  a more diverse comparisons

-> In this study, the reasons for using oriental medicine were analyzed only socially, politically, economically, and scientifically. Through the reference materials you recommended, we were able to check the validity of KM selection based on MD satisfaction. References are also a good source of evidence, but unfortunately, in this study, it is difficult to compare them as references because the satisfaction of MD has not yet been confirmed. Thanks for the good comments.

eg  Kim, D., Lim, B. & Kim, C. Relationship between patient satisfaction with medical doctors and the use of traditional Korean medicine in Korea. BMC Complement Altern Med 15, 355 (2015). https://doi.org/10.1186/s12906-015-0879-x

Reviewer 3 Report

Overall, an outstanding manuscript is submitted. It comprehensively covers the scope of the journal. English text is acceptable. The tables are well structured and informative. Discussions are appropriate and connected to results. Conclusions are given in form of benefits and must be revised.

To be accepted after revision of conclusions.

Author Response

Thank you for your review.
Corrected the result.

Reviewer 4 Report

Review report 

The manuscript discussed the “relationship between the Korean Medicine and Western Medicine in Medical service use. I have some concerns with the paper. One main concern is that partaking both Western medicine and Herbal medicine could lead to liver problems. It is bit confusing about the statistical models used by the authors. The authors should explain this in the introduction. The data could be presented in an effective manner.

Specific comments

·      Are only older people turn to Korean Medicine? 

·      One factor should be price of the medical system.

·      Dose selecting medical system depend on the gender.

·      Dose it also depends on the access to the system.

Author Response

The manuscript discussed the “relationship between the Korean Medicine and Western Medicine in Medical service use". I have some concerns with the paper. One main concern is that partaking both Western medicine and Herbal medicine could lead to liver problems.

-> KM includes Non Herb Medicine (Non-HM) such as acupuncture, cupping and moxibustion. In the case of outpatient medical treatment in Korea's KM field, Non-HM is usually used a lot (Table 3). It would be good to study this risk in future studies by investigating outpatients who mainly use Herb Medicine.

It is bit confusing about the statistical models used by the authors. The authors should explain this in the introduction. The data could be presented in an effective manner.

-> Please see the attachment

Specific comments

  • Are only older people turn to Korean Medicine? -> Respondents those over 65 years of age accounted for the highest proportion of users of KM and WM. (Table 4)
  • One factor should be price of the medical system -> Mean Medical expenses per visit of KM and WM is similar. (Table 3)
  • Dose selecting medical system depend on the gender. -> It was also confirmed from the analysis results that women use outpatient care more overall than men. However, the authors believe that the reason for this is women's willingness to respond to treatment, and it seems difficult to analyze a clear reason.
  • Dose it also depends on the access to the system. -> 

    Accessibility by medical institution was not investigated in the survey used in this study. The accessibility of KM and WM in Korea's medical system is good. Therefore, the authors believe that accessibility is unlikely to have a significant impact on health care use.

    Thanks for the good comments.

Round 2

Reviewer 1 Report

 Definitions of key terms WM, KM, CT  are missing. These key terms must be in the introduction.

Author Response

Definitions of key terms WM, KM, CT  are missing. These key terms must be in the introduction.

-> Thank you for your review. Corrected the introduce and please see the attachment

Reviewer 4 Report

The athuros made the required changes. 

Author Response

Thanks for the review